# Fecal Microbiome Composition Correlates with Pathologic Complete Response in Patients with Operable Esophageal Cancer Treated with Combined Chemoradiotherapy and Immunotherapy

**DOI:** 10.3390/cancers16213644

**Published:** 2024-10-29

**Authors:** Fyza Y. Shaikh, Seoho Lee, James R. White, Yujie Zhao, Jacqueline T. Ferri, Gavin Pereira, Blair V. Landon, Suqi Ke, Chen Hu, Josephine L. Feliciano, Russell K. Hales, K. Ranh Voong, Richard J. Battafarano, Stephen C. Yang, Stephen Broderick, Jinny Ha, Elizabeth Thompson, Eun J. Shin, David L. Bartlett, Benny Weksler, Drew M. Pardoll, Valsamo Anagnostou, Vincent K. Lam, Ali H. Zaidi, Ronan J. Kelly, Cynthia L. Sears

**Affiliations:** 1Department of Oncology, Johns Hopkins University School of Medicine, Baltimore, MD 21205, USA; 2The Sidney Kimmel Comprehensive Cancer Center, Johns Hopkins University School of Medicine, Baltimore, MD 21205, USA; 3The Bloomberg-Kimmel Institute of Cancer Immunotherapy, Johns Hopkins University School of Medicine, Baltimore, MD 21205, USA; 4Department of Medicine, Johns Hopkins University School of Medicine, Baltimore, MD 21205, USA; 5Resphera Biosciences, Baltimore, MD 21231, USA; 6Department of Biomedical Engineering, Johns Hopkins University Whiting School of Engineering, Baltimore, MD 21218, USA; 7Department of Biostatistics, Bloomberg School of Public Health, Johns Hopkins University, Baltimore, MD 21205, USA; 8Department of Radiation Oncology, Johns Hopkins University School of Medicine, Baltimore, MD 21205, USA; 9Department of Surgery, Johns Hopkins University School of Medicine, Baltimore, MD 21205, USA; 10Department of Pathology, Johns Hopkins University School of Medicine, Baltimore, MD 21205, USA; 11Allegheny Health Network Cancer Institute, Allegheny Health Network, Pittsburgh, PA 15212, USA; 12The Charles A. Sammons Cancer Center, Baylor University Medical Center, Dallas, TX 75246, USA

**Keywords:** fecal microbiome, immune checkpoint inhibitors, neoadjuvant chemoradiotherapy, esophageal cancer

## Abstract

This study investigates the fecal and tumor microbiome as potential biomarkers of tumor response to neoadjuvant immunotherapy in combination with chemoradiation. We performed 16S rRNA amplicon sequencing on fecal and tumor samples, as well as fecal metabolomics, on biosamples collected from patients with resectable esophageal or gastroesophageal junction carcinoma before and during treatment. The fecal microbiome of patients whose tumors achieved a pathologic complete response revealed enrichment in sphingolipid and primary bile acids, with corresponding elevated abundance of several bacterial species: *Roseburis inulinivorans*, *Ruminococcus callidus*, and *Fusicantenibacter saccharivorans.* While these results provide initial insight, further study is necessary to validate these data and determine how microbes alter the immune response with neoadjuvant immunotherapy.

## 1. Introduction

Esophageal cancer constitutes a significant contributor to global cancer-related fatalities, with an estimated incidence of 21,000 cases and 16,000 deaths in 2023 [1]. Current guidelines for stage II or III esophageal and gastroesophageal cancer recommend neoadjuvant chemoradiotherapy (CRT) followed by surgery. However, the majority of patients will not achieve a pathologic complete response (PCR) at the time of surgical resection, and residual viable tumor (RVT) at surgical resection is associated with lower overall survival. For patients with RVT, recent advancements have led to the FDA approval of adjuvant nivolumab based on improved disease-free survival [2,3,4]. While the ICI contribution has been validated clinically, the addition of immune checkpoint inhibitors (ICIs) to neoadjuvant regimens in esophageal cancer or as part of a perioperative approach with chemotherapy remains under investigation. Neoadjuvant ICIs have shifted the treatment paradigm in lung and breast cancer, as well as melanoma and Merkel-cell carcinoma, and pathological response is emerging as a surrogate marker for overall survival [5]. However, whether PCR can be used to determine long-term efficacy in esophagogastric cancers remains unclear, and multiple phase 3 studies have been conducted in operable esophageal and gastroesophageal cancer, to evaluate efficacy of neoadjuvant immunotherapy with chemotherapy and/or CRT [6,7,8,9,10].

While pathologic response has correlated with outcomes, the evaluation of tissue remains limited to the time of resection. The development of additional biomarkers in addition to PD-L1, MSI-H/dMMR, and HER2 to guide treatment is essential to maximize curative potential for patients [11]. Several tumor and fecal microbiome features have been shown to correlate with esophageal tumors and ICI response, and specifically, with pathologic response [12,13]. Additional studies have examined microbiome features with neoadjuvant CRT, identifying enrichment of Fusobacteriaceae in late-stage disease and Lactobacillaceae associated with partial and complete response [14] as well as post-surgical alterations in the gut microbiome [15]. Moreover, intratumoral *Fusobacterium nucleatum* has been shown to be enriched in ICI nonresponders, and high intratumoral levels correlate with poor response to neoadjuvant chemotherapy [16,17,18]. In general, there is significant variability in associations between microbial genus and species and ICI response, and this lack of consensus creates challenges in using microbiome taxonomic analysis as a biomarker to guide immunotherapy.

It has also been suggested that microbial metabolites may be the key mediators of cancer immunity and ICI responsiveness that could enhance biomarker and interventional strategies to cancer immunotherapy [19]. A number of metabolites produced by or linked to microbes, such as short chain fatty acids, inosine, and trimethylamine N-oxide, have been implicated in other tumor types as modulators of immune cell function in the setting of ICI treatment [20,21,22,23]. In esophageal cancer, the specific microbial metabolites that may influence anti-tumor immunity remain unknown, either in the colon or in the tumor microenvironment

Herein, we present correlative taxonomic and metabolomic microbiome biomarkers for a subset of patients who received neoadjuvant nivolumab or nivolumab–relatlimab combined with CRT in patients with operable stage II/III E/GEJ (NCT03044613) [24]. We hypothesize that microbial taxa and/or metabolic pathways correlate with tumor response to neoadjuvant immunotherapy. Our data show that patients with PCR after ICI treatment have a distinct fecal microbiome profile compared to those with RVT, as well as including uniquely enriched fecal metabolites that correlate with individual bacterial species. These data have the potential to further distinguish pathologic responses in esophageal or gastroesophageal cancer in addition to tissue biomarkers at the time of surgical resection.

## 2. Methods

### 2.1. Study Design, Eligibility Criteria, and Participants

All participants were enrolled in an open-label, multi-institution study enrolling patients at Johns Hopkins Sidney Kimmel Cancer Comprehensive Cancer Center in Baltimore, MD, USA, Allegheny Health Network in Pittsburgh, PA, USA, and Baylor University Medical Center in Dallas, TX, USA (NCT03044613). This protocol consisted of a phase IB trial of induction nivolumab or nivolumab/relatlimab prior to concurrent chemoradiation (weekly carboplatin/paclitaxel and concurrent radiation) [24]. The study protocol and all amendments were approved by the Institutional Review Board of Johns Hopkins University (Johns Hopkins Medicine Institutional Review Board #6) and local institutions (Allegheny Singer Research Institute and Baylor Scott & White Research Institute). Written informed consent was obtained from all participants; all patients were enrolled from 23 August 2017 to 1 July 2021 with data lock on 25 January 2022. Full eligibility, exclusion, enrollment, and outcome data have been previously published [24].

### 2.2. Fecal Sample Collection

All fecal samples were collected by patients at home as either fresh stool or placed into an OMNIgene GUT (DNA Genotek, Stittsville, ON, USA, OM-200). Fresh stool was stored at 4 °C, brought to the clinic within 48 h, and aliquoted in a laminar flow hood. OMNIgene GUT tubes were mailed or brought to the lab within 60 days, vortexed, and aliquoted in an anaerobic hood. All samples were stored at −80 °C.

### 2.3. DNA Extraction and Sequencing

For fecal samples, approximately 80 mg of fresh stool or 250 uL of OMNIgene GUT stool sample were transferred to BashingBead Lysis Tubes (Zymo, 0.1/0.5 mm beads) with bead bashing buffer, followed by mechanical lysis using a Mini-Beadbeater-96 (Biospec Products, Bartlesville, OK, USA) at 2400 rpm for 60 s, 3 cycles in total. The remainder of the DNA extraction was performed using the Quick-DNA Fecal/Soil Microbe Kit (Zymo, Irvine, CA, USA, 96 well plate). For tissue samples, DNA quantity and purity were measured using a Nanodrop (Thermo Scientific, Waltham, MA, USA). All sequencing was performed by the CHOP Microbiome Center as previously described [25].

### 2.4. 16S rRNA Data Analysis and Statistics

Alpha and beta diversity was calculated using QIIME (v.1.8.0) to generate Bray–Curtis dissimilarity scores between samples. Statistical analysis was performed using permutational multivariate analysis of variance. Pairwise comparisons were performed at all taxonomic levels using a Wilcoxon rank-sum test and corrected for false discovery rate (FDR, *p* < 0.05). Microbiome feature discovery and effect size was performed using the linear discriminant analysis effect size (LEfSe) method [26]. Comparisons across > 2 groups were performed using a Kruskal–Wallis test. Ambiguous OTUs were excluded from final visualizations but included in all analyses and Appendix A.

### 2.5. Fecal Metabolite Extraction, Data Acquisition, and Analysis

For each sample, 80 mg of fecal material was transferred to a sterile tube with a 3 mm metal bead (Qiagen, Hilden, Germany) in a laminar flow hood on ice. Metabolites were extracted in ice-cold 80% methanol with vortex homogenization. Prior to analysis, the extracts were diluted 1:225 with 80% (*v*/*v*) methanol in water. Metabolome profiles were acquired using flow injection mass spectrometry by General Metabolics (Cambridge, MA, USA) [27], with 5 uL of each sample being injected twice, consecutively, within 0.96 min to serve as technical replicates. A pooled study sample (pSS) was prepared by pooling 5 uL of each sample together and was injected periodically throughout the batch. The samples were acquired randomly within the plates. Raw profile data were centroided, merged, and recalibrated using MATLAB software (https://www.mathworks.com/products/matlab.html, accessed on 31 August 2023). Putative annotations were generated based on compounds contained in the Human Metabolome Database, KEGG, and ChEBI databases using both accurate mass per charge (tolerance 0.001 *m*/*z*) and isotopic correlation patterns. To identify significant metabolites enriched in PCR vs. RVT groups, mean and variance differential analysis were performed comparing each group and displayed using R (v4.3.3), tidyverse (v2.0.0), rstatix (v0.7.2), plotly (v4.10.4), ggrepel (v0.9.5), reshape2 (v1.4.4), and ggpubr (v0.6.0). Ions with *p*-value ≤ 0.05 and |log2 Fold Change| ≥ 1 were considered significant. For pairwise comparisons, significance was assessed using a Wilcoxon ranksum test.

## 3. Results

### 3.1. Patient Characteristics and Pathologic Response

A total of 42 patients were screened for the trial and 32 patients were enrolled [24]. A subset of these patients (*n* = 23) was able to provide analyzable fecal samples within the first 42 days of treatment (Table 1). Patients were treated with neoadjuvant nivolumab (Arm A, *n* = 11) or nivolumab–relatlimab (Arm B, *n* = 12) in combination with chemoradiotherapy followed by surgical resection. Of note, following a protocol amendment for arm B2, nivolumab–relatlimab was only given as induction therapy prior to combination chemoradiotherapy due to high-grade toxicity when combined with CRT (arm B1) as previously described [24]. On pathologic evaluation, 35% (*n* = 8) had a PCR while 65% (*n* = 15) had residual viable tumor (RVT). Among the patients with RVT, six had a major pathologic response (MPR or ≤10% RVT) while nine had >10% RVT after neoadjuvant treatment. A full sample list with clinical and pathological characteristics is shown in Appendix A and microbiome features for each sample are shown in Appendix A (multiple taxonomic levels) with functional prediction (Appendix A).

### 3.2. Fecal Microbiome Features Correlate with Pathologic Response After Neoadjuvant Treatment

The first two fecal samples from each patient (both pre-treatment and on-treatment) were analyzed for microbiome composition by 16S rRNA amplicon sequencing using V1V2 primers using a high-resolution taxonomic approach as previously described [25], and the overall microbiome composition was analyzed using Bray–Curtis, a metric for beta diversity that compares distinct communities. The overall microbiome composition using the first two fecal samples was distinct between patients with subsequent PCR vs. RVT (Figure 1A), and the association with tumor response was not affected by correction for study arm (*p* = 0.0070). When separating RVT into those with 10% RVT (MPR) vs. >10% RVT, the fecal microbiome profiles also remained distinct based on tumor response (Figure 1B).

To determine unique features, fecal samples from patients with PCR tumors vs. RVTs were analyzed for features distinguishing the two groups. Using the 10 most abundant genera across all groups (Figure 1C), a linear discriminate analysis (LDA) showed that these features alone were sufficient to distinguish PCR vs. RVT groups (Figure 1D, Appendix A). Next, to identify species enriched in each group, two distinct approaches were used. Specifically, microbiome species enriched in each group were analyzed using both a pairwise statistical approach with correction for false discovery rate (FDR, Figure 1E, Appendix A) as well as linear discriminant analysis effect size (LEfSe, Figure 1F, Appendix A). Several species were identified by both approaches: *Ruminococcus callidus*, *Fusicatenibacter saccharivorans*, and *Roseburia inulinivorans* as enriched in PCR *(p* < 0.001). Of note, several operational taxonomic units (OTUs) can be identified for each species using 16S amplicon sequencing data, and for *R. inulinivorans*, the primary OTU appears to correlate with PCR while a second OTU may correlate with RVT, though the relative abundance is lower (Figure 1F, Appendix A). LEfSe was also utilized to assess all taxonomic levels, which confirmed the analysis performed at the species level (Appendix A, Appendix A).

### 3.3. Fecal Metabolic Profiles Associated with Pathologic Response Correlate with Microbiome Features

In order to explore the function of the gut microbiome, available samples were analyzed by global untargeted metabolomics and analyzed for features associated with pathologic response (Appendix A, *n* = 22 patients, *n* = 43 fecal samples). Several metabolites were found enriched in the PCR vs. RVT group (Figure 2A). Of these, C16-ceramide (C16Cer, Figure 2B) and chenodeoxycholic acid (CDCA, Figure 2D) have been implicated in the microbial metabolism. Several gut bacteria, notably from the *Bacteroides* genus, are known to produce sphingolipids which are processed by host metabolic pathways [28]. Thus, we examined the abundance of *Bacteroides* relative to tumor response and found that its abundance correlated with improved tumor response (Figure 2C). Additionally, we correlated all enriched metabolites (from Figure 2A) with all enriched species (from Figure 1E,F) using a Spearman correlation (Figure 2E, Appendix A) to determine if the relative abundance of a particular species correlated with a change in the abundance of significant metabolites. C16Cer and chenodeoxycholic acid most strongly correlated in abundance with *R. callidus* (*p* < 0.01), in addition to other bacterial species. Notably, however, this analysis is only correlative with metabolites and OTUs that were demonstrated to be significant by other tests, and thus, the results should be considered exploratory.

### 3.4. Tumor Microbiome Features Associated with Pathologic Response

Previous work has shown that specific microbes have been associated with poor outcomes to neoadjuvant chemotherapy in esophageal cancer, notably *Fusobacterium nucleatum*, while other microbial taxa have been associated with improved responses to ICIs (i.e., *Streptococcus*) [13,16,17,18]. As an exploratory analysis due to limited sample size, we performed 16S rRNA amplicon sequencing on tumor DNA to determine the presence of microbes in tumors (*n* = 12 patients, *n* = 12 samples) with paired normal tissue samples as controls when available (*n* = 8 patients, *n* = 8 samples, Appendix A). The top genera are shown for all tumor samples, grouped by pathologic response (Figure 3A), and differential abundance analysis by pathologic response, revealed the genus *Veillonella* and species *Streptococcus mitis* as enriched in RVTs (*p* < 0.05, Figure 3B,C) while *Collinsella aerofaciens* was enriched in PCR tumors (*p* < 0.05, Figure 3D). *F. nucleatum* was detected in only 1 of 12 tumors in a patient with MPR (Appendix A).

## 4. Conclusions

While the microbiome has been associated with outcomes in the metastatic setting in esophageal cancer, limited data exist on microbiome features associated with pathologic tumor response to ICIs in the neoadjuvant setting. Herein, we present data revealing that the microbiome profiles of patients with a PCR after neoadjuvant PD-1/LAG-3 are distinct, both taxonomically and metabolically, from those with RVT on pathologic evaluation. Strikingly, the fecal microbiome composition can be distinguished using only the genus level with the 10 most abundant genera across all samples. Further, these data identify specific species and metabolites that are enriched in the PCR group, and these findings point to interesting biological functions that warrant further exploration to identify additional pathways that might be amenable to additional targeted therapy.

Among our key findings, we identified several *Bacteroides* species as enriched in the PCR group. As mentioned in the results, *Bacteroides* species are producers of sphingolipids, which can impact host metabolism and regulate cholesterol metabolism [28]. In fact, bacterial sphingolipids have been shown to be incorporated into the host liver and even rescue excess lipid accumulation in a mouse model of hepatic steatosis [29]. While the exact contribution of bacterial vs. dietary sphingolipids remains under investigation, the modification of liver function could impact treatment efficacy or toxicity for systemic chemo- or immunotherapies for cancer patients. Second, the PCR group also showed enrichment of a primary bile acid, CDCA, which is synthesized in the liver and largely reabsorbed in the small intestine [30]. When unabsorbed, CDCA or its bacterially derived conjugates (i.e., secondary bile acids) can act as signaling molecules by stimulating the farsenoid X receptor (FXR). FXR regulates bile acid reabsorption, but the receptor also has multiple roles in cell signaling, including promoting an anti-inflammatory effect through downregulation of TNFα, IL-1, and IL-6 as well as strengthening the integrity of the epithelial barrier, which serves as an antibacterial defense [31]. Higher fecal levels of secondary bile acids have also been linked to ICI response in hepatocellular carcinoma [32].

In addition to the fecal microbiome, we performed an exploratory analysis on the tumor microbiome. Prior work in the field has identified several species whose increased abundance has been found in esophageal tumors. Notably, these include *Prevotella*, *Fusobacterium*, and *Streptococcus* species [33,34]. A recent study found that *Streptococcus* specifically could be used as a marker of disease-free survival as the abundance positively correlated with the infiltration of CD8^+^ T-cells into the tumor microenvironment [13]. In our data, *Streptococcus mitis* is more abundant in the RVT group (Figure 3C). The variation in microbial detection among tumors could be due to a variety of factors, including geographical sampling, diet, and redundant biological functions between bacterial species. Given our small sample size, we were unfortunately unable to perform extensive analysis or FDR correction. Additional analysis was also limited by detection, as the genus *Prevotella* was detected in 7 of 12 tumors without clear enrichment in either group, *Fusobacterium nucleatum* in 1 of 12 tumors (3 of 8 paired normal tissue), and *Fusobacterium periodonticum* in only 1 of 12 tumors (2 of 8 paired normal tissue).

While these findings are promising, our study has several limitations. First, this is a small cohort study evaluating patients on a neoadjuvant immunotherapy clinical trial. However, recent data from the phase III ESOPEC trial indicate that perioperative chemotherapy may outperform CRT, with 19.3% (95%-CI 13.9–25.9%) of participants achieving pathologic complete response in the perioperative chemotherapy arm, compared to 13.5% (95%-CI 8.8–19.4%) in the conventional CRT arm based on the CROSS regimen (*n* = 359 patients resected) [35,36]. Our hypothesis and rationale for immunotherapy use in the neoadjuvant setting includes the potential for tumor-associated antigens to prime the immune response to prevent recurrence [37]. The long-term data for overall survival and recurrence-free survival will need to be compared across multiple studies, pending maturity, to determine if immunotherapy has an added benefit for patients with resectable solid tumors.

Additional limitations include sample size and lack of data on microbiome modifiers. First, given that we are assessing the microbiome features with neoadjuvant immunotherapy in the clinical trial setting, our sample size remains limited and may not be comparable to existing data in the field in the metastatic setting. Our study uses 16S rRNA amplicon sequencing data, which is limited to taxonomic identification and does not identify other genetic markers in the microbial community. Thus, we utilized a metabolite-based screen to assess functional markers. Second, this study enrolled patients across three geographical sites in the United States (Baltimore, Dallas, and Pittsburgh) and most previously published data are based on studies in East Asia. These geographical differences, with associated diet and medication variables, can contribute significant variation to the baseline microbiome and thus the fecal or tumor microbiome in cancer patients. Finally, both the taxonomic and metabolomic findings are correlative with clinical outcomes and require further validation in a prospective cohort. Moreover, detailed mechanistic studies are still required to understand the underlying mechanism(s) and host–microbe interactions with microbial metabolites and how those interactions ultimately impact tumor biology and response to treatment.

In summary, these data show that the fecal microbiome composition and the functional components through metabolite production are distinct in patients with operable esophageal cancer who have a PCR to neoadjuvant combination ICI and CRT. These data, with additional validation, may potentially be extrapolated to guide treatment in additional neoadjuvant studies and/or define tumor response in the metastatic setting.

## Figures and Tables

**Figure 1 cancers-16-03644-f001:**
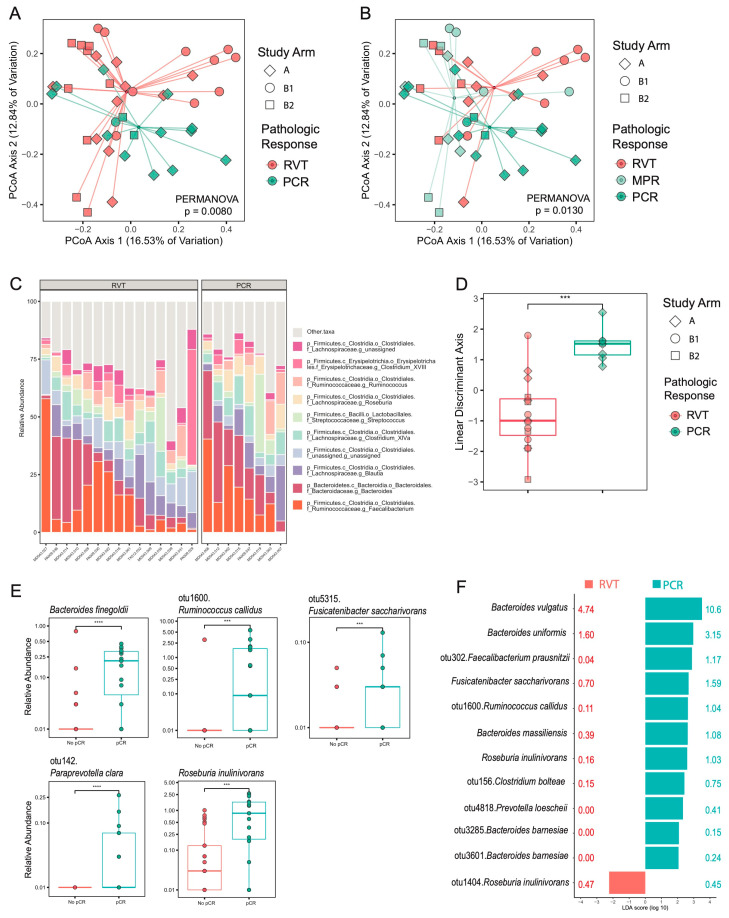
Fecal microbiome features correlate with pathologic response. (**A**) Principal coordinates analysis (PCoA) using the Bray–Curtis distance metric by pathologic complete response (PCR) compared to those with residual viable tumor (RVT). Shape represents study arm while color represents tumor response group. Statistics: permutational multivariate analysis of variance (PERMANOVA). (**B**) PCoA using the Bray–Curtis distance by pathologic complete response (PCR), major pathologic response (MPR, <10% RVT), and RVT > 10%. Statistics: PERMANOVA (**C**) stacked histogram representing the relative abundance of the top 10 most abundance genera in fecal samples by tumor response group. (**D**) Linear discriminate analysis (LDA) using the relative abundance of the top 10 most abundance genera by response group. Statistics: Mann–Whitney. (**E**) Pairwise comparisons of the bacterial species that are significant between tumor response groups. Statistics: Mann–Whitney with correction for false discovery rate (FDR, *p* < 0.05). (**F**) Significant features at the species level that are mostly likely to explain differences between tumor response groups using linear discriminant analysis effect size (LEfSe). The direction of the bar and the color indicate enrichment in tumor response group. The value, by color, next to the bar represents the mean abundance in each tumor response group (Appendix A). *** *p* < 0.001, **** *p* < 0.0001.

**Figure 2 cancers-16-03644-f002:**
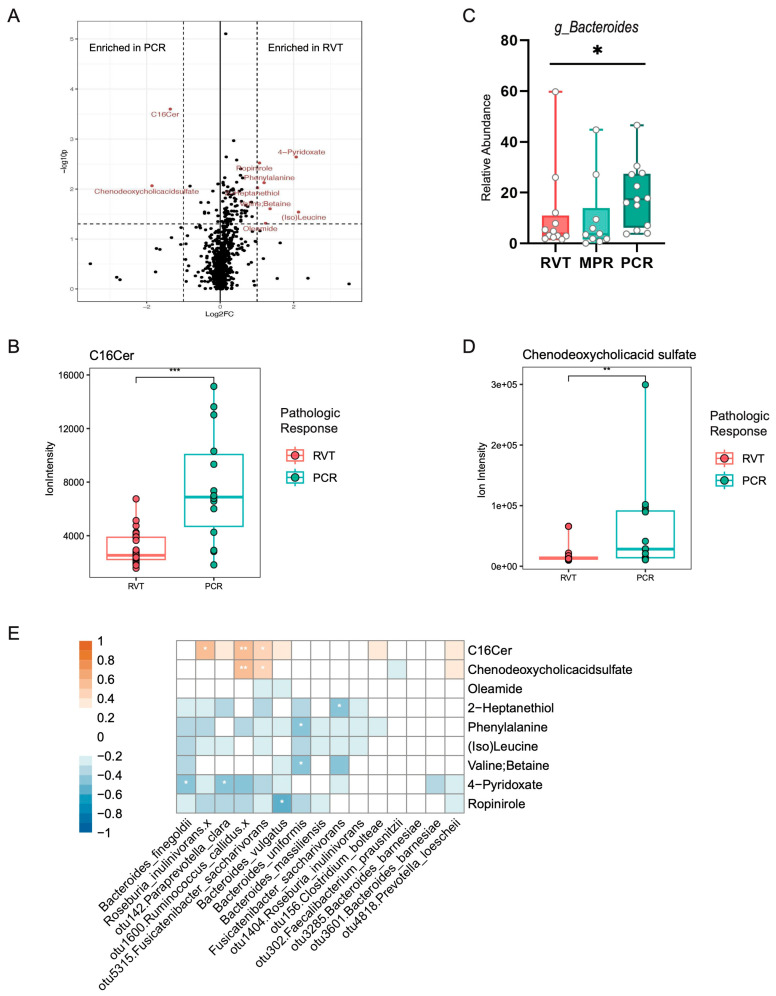
Fecal metabolic profiles associated with pathologic response correlate with microbiome features. (**A**) Volcano plot showing significant metabolites enriched in PCR or RVT groups by |log2FC| ≥ 1 and *p*-value ≤ 0.05. (**B**) Ion intensity for C16-ceramide (C16Cer) by tumor response group. Statistics: Mann–Whitney. (**C**) Relative abundance of *Bacteroides* genus by tumor response groups. Statistics: Kruskal–Wallis test. (**D**). Ion intensity for Chenodeoxycholic acid (CDCA) sulfate by tumor response group. Statistics: Mann–Whitney. (**E**) Spearman correlation of all enriched metabolites (from Figure 2A) with all enriched species (from Figure 1E,F). Red indicates positive correlation; blue indicates negative correlation. Statistics: FDR-corrected, *, *p* < 0.05; ** *p* < 0.01, *** *p* < 0.001.

**Figure 3 cancers-16-03644-f003:**
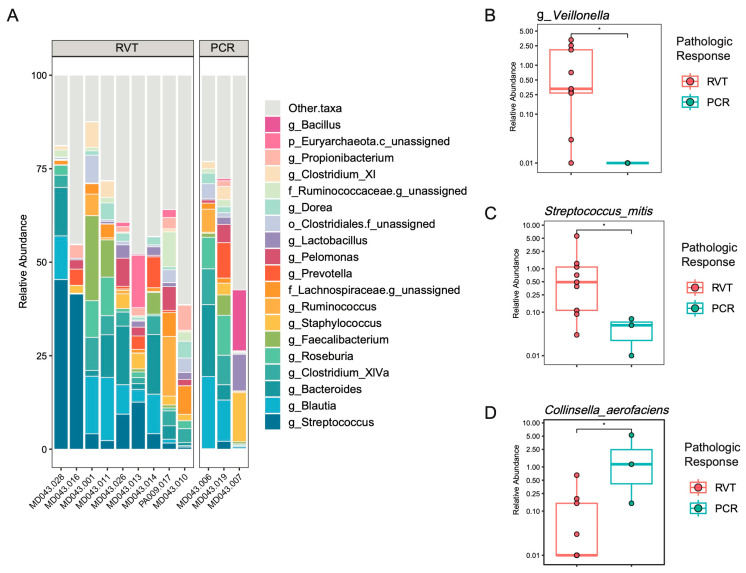
The tumor microbiome features associated with pathologic response. (**A**) A stacked histogram representing the relative abundance of the top 20 most abundance taxa in tumor samples by tumor response group. (**B**–**D**) Significant genus and species level taxa by pairwise comparisons of tumor response groups. Statistics: Mann–Whitney. * *p* < 0.05.

**Table 1 cancers-16-03644-t001:** Patient characteristics by tumor response groups. Abbreviations. PCR, pathologic complete response; RVT, residual viable tumor; SD, standard deviation; CAP, College of American Pathologists.

Characteristic	PCR, *n* = 8 ^1^	RVT, *n* = 15 ^1^	*p*-Value ^2^
**Study Arm (%)**			0.3
A	6 (75%)	5 (33%)	
B1	1 (13%)	5 (33%)	
B2	1 (13%)	5 (33%)	
**Sex at Birth (%)**			>0.9
Female	1 (13%)	1 (6.7%)	
Male	7 (88%)	14 (93%)	
**Age [yrs] (SD)**	62 (5)	65 (6)	0.3
**Smoking History (%)**			0.089
Former	6 (75%)	5 (33%)	
Never	2 (25%)	10 (67%)	
**Tumor Histology (%)**			>0.9
Adenocarcinoma	7 (88%)	13 (87%)	
Squamous	1 (13%)	2 (13%)	
**CAP Regression Score (%)**			<0.001
G0	8 (100%)	0 (0%)	
G1	0 (0%)	6 (40%)	
G2	0 (0%)	8 (53%)	
G3	0 (0%)	1 (6.7%)	
**Residual Viable Tumor at Resection (%)**			<0.001
0%	8 (100%)	0 (0%)	
<10%	0 (0%)	6 (40%)	
>10%	0 (0%)	9 (60%)	

^1^ *n* (%); Mean (SD). ^2^ Fisher’s exact test; Wilcoxon rank sum test.

## Data Availability

Authors consent to the sharing of data and materials. Raw sequencing data are available (NCBI PRJNA1142503). The code to process the raw sequencing data, analysis, and figure generation is available at https://github.com/fshaikh14/shaikh-sears-esca-microbiome-neoadj-2024 (accessed on 20 October 2024).

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
