# Peer review of "Fecal Microbiome Composition Correlates with Pathologic Complete Response in Patients with Operable Esophageal Cancer Treated with Combined Chemoradiotherapy and Immunotherapy"

_cancers, 2024, doi:10.3390/cancers16213644_

Round 1
Reviewer 1 Report
Comments and Suggestions for Authors
The manuscript provides insight into characteristics of the esophageal cancer treated by
Combination of chemoradiotherapy and immunotherapy. The characteristics were obtained by 2 different technologies, 16S (tumor and gut) and flow-injection mass spectrometry (gut) and this combination are not available in the literature. The authors make available all obtained results in the supplementary tables. The data that can be used for comparison with other tumors and for validation of new hypothesis.
I believe that study should be published, but I would recommend the major revision to improve presentation and organization of the results.
In general, the Introduction gives a nice overview of recent publications on neoadjuvant CRT and ICI, but not on what is known on interaction between microbes in the gut and in the tumor and what metabolites produced in the gut can be linked to specific microbes and to response to CRT and ICI. Some statements need references, such as “However, 70-75% of patients will 54 not achieve a pathologic complete response (PCR) at the time of surgical resection”, and “It has been suggested that microbial metabolites may 81 be the key mediators of cancer immunity and ICI responsiveness that could enhance bi-82 biomarker and interventional strategies to cancer immunotherapy.”
There is no clear statement of hypothesis in the introduction.
Method section.
1. Some additional information maybe helpful.
2. It will be nice if authors explain why 10% threshold was used to categorize the response,
3. What specifically was used for CRTand ICI?
4. I believe that LEfSe should be redone by correcting input for the analysis.
5. Quality of figures is very low. Name of species and some numbers are not recognizable.
Results
I would recommend improving organization of the supplementary data.
It would be nice to see titles in each Supplementary tables, units, and meanings of the abbreviations that are not straightforward. Some tables, such as Table S1 is huge with different types of data grouped together. It was difficult to find information in the table. It would be nice if authors structured the table according to different data types, maybe by transposing the table.
Major concerns :
1. Three arms of the study, A,B, and B2 have very different outcome in terms of response (See Table 1) and tend to be different in the major phyla abundances in the gut according to ttest. I noticed that Firmicutes are more abundant in arm B+B2. vs A, while Bacteroidetes and Actinobacteria are more abundant in arm A vs B+B2. I am afraid that the difference introduces bias in the comparison of PCR and RVT when all arms are combine. A separate comparison may give different results. I think it is important to show that it is now the case.
2. The search of taxa discriminating the pathologic response in the study was implemented only at the species level, and identified spp., are not very abundant and often rare.
The LEfSe tool provides an opportunity to characterize taxa that are differentially abundant between 2 conditions at different taxonomic levels, from species to phyla and visualize the results as cladograms. I believe that it is the correct way to analyze data in the study. It will only require reformatting input table for the tool.
3. I am not sure I see statistical evidence for the statement “Fecal metabolic profiles associated with pathologic response correlate with microbiome features.”. Both, microbial features and metabolites are selected for the correlation analysis because they are either enriched or depleted in the same 2 groups: PCAsamples and RVT-samples. Therefore, It is not surprising that they are correlated. The problem is that false discovery of the correlation might be high. I would suggest focusing on biology and direction of the identified correlations, like you did for Bacteroides. Namely, I would recommend, in addition to Fig.2E, to create a summary table or Fig. to show metabolites, gut bacteria, and intratumoral bacteria enriched in each condition PCR and RVT. The table will integrate major results obtained by screening gut microbes, gut metabolites, and tumor microbes. You can show arrows between metabolites and bacteria, and bacteria in the gut and withing the tumor, if you find evidence related to the association in the literature. I believe the summary Figure will be important to grasp the results and discuss them.
5. At this point the section on Intratumoral microbiome in esophageal cancer is not connected with other results. It is not clear how Intratumoral microbiome in esophageal cancer is different from normal tissue. I didn’t see any conclusions about that. As I have mentioned above, one way to integrate intratumoral microbes that effect the treatmenr response with those obtained for the gut and with meabolites, can be a summary Figure. I believe that the summary table will be also helpful to formulate the major hypothesis of the study in the introduction to glue sections of the manuscript.
Author Response
Please see attached word document with detailed response to all editor and reviewer comments.

Reviewer 2 Report
Comments and Suggestions for Authors
The manuscript in case focuses on the identification of fecal microbiome features and metabolites that correlate with treatment outcomes, as well as the analysis of microbiome profiles within tumor tissues. An exploratory analysis of tumor tissues using 16S rRNA amplicon sequencing revealed significant differences in microbiome composition between PCR and RVT tumors. The authors observed distinct microbiome profiles and metabolomic patterns between PCR and RVT patients suggesting that gut bacteria and their metabolites may play a critical role in influencing treatment outcomes. While associations are observed, it is unclear whether specific microbiome features or metabolites actively contribute to the treatment response or are merely bystanders. Without functional experiments or mechanistic studies, the causative role of the identified bacterial species or metabolites remains speculative. The study does not extensively control for potential confounding factors such as antibiotic use, proton pump inhibitors, diet, and other medications, which are known to influence the gut microbiome composition.
Additionally:
- 16S rRNA sequencing cannot capture the full extent of microbial diversity or provide detailed information on microbial functions. This makes it difficult to draw firm conclusions about the specific species or metabolic pathways that might be involved in modulating treatment responses.
- Keywords: neoadjuvant chemoradiotherapy is more appropriate
- Abstract, line 30: I consider it's just the chemoradiotherapy that has the potential to prime the immune system for the optimal response to immunotherapy/check-point inhibitors
- Introduction, line 56: please provide reference
- Results: why wasn't a chemotherapy only arm? this is worth discussing as the recently published (June) ESOPEC trial revealed that perioperative FLOT chemotherapy outperformed neoadjuvant CRT
Author Response
Please see attached word document for response to all editor and reviewer comments.

Round 2
Reviewer 1 Report
Comments and Suggestions for Authors
I believe that the paper can be accepted for publication.